# Lymphocyte deficiency alters the transcriptomes of oligodendrocytes, but not astrocytes or microglia

**Mitchell C. Krawczyk[1], Lin Pan[1], Alice J. Zhang[1], Ye Zhang**[1,2,3,4]*

**1** Department of Psychiatry and Biobehavioral Sciences, Intellectual and Developmental Disabilities Research Center, Semel Institute for Neuroscience and Human Behavior, David Geffen School of Medicine, University of California Los Angeles (UCLA), Los Angeles, Los Angeles, California, United States of America, **2** Brain Research Institute, University of California Los Angeles (UCLA), Los Angeles, Los Angeles, California, United States of America, **3** Eli and Edythe Broad Center of Regenerative Medicine and Stem Cell Research, University of California Los Angeles (UCLA), Los Angeles, Los Angeles, California, United States of America, **4** Molecular Biology Institute, University of California Los Angeles (UCLA), Los Angeles, Los Angeles, California, United States of America

* yezhang@ucla.edu

## Abstract

Though the brain was long characterized as an immune-privileged organ, findings in recent years have shown extensive communications between the brain and peripheral immune cells. We now know that alterations in the peripheral immune system can affect the behavioral outputs of the central nervous system, but we do not know which brain cells are affected by the presence of peripheral immune cells. Glial cells including microglia, astrocytes, oligodendrocytes, and oligodendrocyte precursor cells (OPCs) are critical for the development and function of the central nervous system. In a wide range of neurological and psychiatric diseases, the glial cell state is influenced by infiltrating peripheral lymphocytes. However, it remains largely unclear whether the development of the molecular phenotypes of glial cells in the healthy brain is regulated by lymphocytes. To answer this question, we acutely purified each type of glial cell from immunodeficient Rag2$^{-/-}$ mice. Interestingly, we found that the transcriptomes of microglia, astrocytes, and OPCs developed normally in Rag2$^{-/-}$ mice without reliance on lymphocytes. In contrast, there are modest transcriptome differences between the oligodendrocytes from Rag2$^{-/-}$ and control mice. Furthermore, the subcellular localization of the RNA-binding protein Quaking, is altered in oligodendrocytes. These results demonstrate that the molecular attributes of glial cells develop largely without influence from lymphocytes and highlight potential interactions between lymphocytes and oligodendrocytes.

## Introduction

The immune system and the nervous system are two vital and intricate biological systems. In recent decades an additional layer in their complexity is emerging as accumulating evidence

**Data Availability Statement:** All RNA-sequencing data from this study are available via the Gene Expression Omnibus, accession number

GSE210580. The outputs of additional analyses are available in the Supporting Information files.

**Funding:** This work is supported by the Achievement Rewards for College Scientists Foundation Los Angeles Founder Chapter and the National Institute of Mental Health of the National Institutes of Health (NIH) Award T32MH073526 to M.C.K, the National Institute of Neurological Disorders and Stroke of the National Institute of Health (NIH) R00NS089780, R01NS109025, the National Institute of Aging of the NIH R03AG065772, the National Institute of Child Health and Human Development P50HD103557, National Center for Advancing Translational Science UCLA CTSI Grant UL1TR001881, the W. M. Keck Foundation Junior Faculty Award, UCLA Eli and Edythe Broad Center of Regenerative Medicine and Stem Cell Research (BSCRC) Innovation Award, the UCLA Jonsson Comprehensive Cancer Center and BSCRC Ablon Scholars Program, and the Friends of the Semel Institute for Neuroscience & Human Behavior Friends Scholar Award to Y. Z. The funders had no role in study design, data collection and analysis, decision to publish, or preparation of the manuscript.

**Competing interests:** I have read the journal's policy and the authors of this manuscript have the following competing interests: Ye Zhang consulted for Ono Pharmaceutical. This does not alter our adherence to PLOS ONE policies on sharing data and materials.

suggests how these two systems interact and influence one another. Although the brain has been traditionally considered an immune-privileged organ, researchers have reported the presence of immune cells in the protective layers surrounding the brain, the meninges, as well as in the perivascular space and choroid plexus [1–4]. Some meningeal immune cells are produced locally in the skull bone marrow and display different properties than immune cells derived from the periphery, suggesting brain-specific roles for the immune cells that occupy this niche [5, 6]. These meningeal immune cells are poised for direct signaling to the brain through secreted factors or indirect signaling via border cells. Several lines of evidence implicate meningeal immune cells in homeostatic brain function. Limiting immune cell migration across the blood-meningeal barrier using VLA-4 integrin antibody results in cognitive impairment, and disrupting meningeal T cells via eliminating the deep cervical lymph nodes also impairs learning [7–9]. One recent study found that meningeal γδ T cells induce anxiety-like behavior through the secretion of IL-17a through activation of receptors on glutamatergic cortical neurons [10]. As the peripheral immune cells influence the brain, the brain in turn regulates the immune system in several ways, including the production of hormones. Activation of the hypothalamic-pituitary-adrenal axis results in the secretion of corticosteroids that inhibit many immune responses [11, 12].

Although the impact of peripheral immune cells on the behavioral output of the central nervous system (CNS) has been demonstrated, how immune cells affect the cellular state in the CNS remains elusive. Glial cells including microglia, astrocytes, oligodendrocytes, and oligodendrocyte precursor cells (OPCs) make up a large portion of brain cells and play key roles in the development and function of the CNS [13–29]. Of particular note in this study, microglia are CNS-resident innate immune cells and key players in CNS pathogen defense, homeostasis, and developmental synapse engulfment and neural circuit refinement [30–39]. Oligodendrocytes form insulating myelin sheaths around axons, accelerate the propagation of action potentials along axons, and provide metabolic support to axons [40–48]. Under pathological conditions in a wide range of neurological disorders, such as stroke, trauma, and CNS autoimmunity, infiltrating peripheral immune cells release cytokines that impact levels of neuroinflammation and glial cell states [49]. State changes of glial cells in turn contribute to neuroinflammation, tissue homeostasis, and neural repair [50]. A long-standing question that remains largely unanswered is whether the cellular states of glial cells are regulated by immune cells under homeostatic conditions in the healthy brain.

Like glia, lymphocytes play a powerful role in many neurological pathologies. Most notably, lymphocytes have been implicated in the prototypic inflammatory disease, multiple sclerosis (MS). Lymphocytes are implicated in the causal pathology of MS due to a variety of experimental observations [51]. Activated myelin-specific T lymphocytes are sufficient to generate brain lesions in the popular mouse model of MS, experimental autoimmune encephalomyelitis (EAE) [52]. Though exceedingly rare in homeostasis, peripheral immune cells can migrate into the brain in a variety of neurological disease states, including stroke, cancer, where they become central players in the pathology [53, 54]. Peripheral myeloid cells and neutrophils can be found in the brains of Alzheimer disease patients, and peripheral immune composition shows changes in Parkinson disease [55, 56]. Inflammation, and therefore immune cells, are known or suspected to play a role in a huge array of CNS diseases. While their many roles in disease garner widespread attention, relatively little is known about how immune cells effect the brain in the absence of disease.

Rag2$^{-/-}$ mice are a particularly useful tool for assessing the impact of peripheral immune cells on the CNS as they lack mature lymphocytes, the central players in adaptive immunity. Lymphocytes, including T and B cells, serve to recognize potentially hazardous antigens, and they accomplish this task by expressing a great diversity of receptors to identify the many

possible antigens they may need to detect. Rather than expressing an impossibly large number of distinct receptor genes, this receptor diversity is accomplished by physical recombination of a small number of antigen receptor genes; Rag1 and Rag2 are the recombinase enzymes required for this recombination [57]. In the absence of Rag2, this recombination cannot take place, so lymphocytes will not create the appropriate receptor array and therefore fail to mature into functional T and B cells [58]. In the absence of these lymphocytes, researchers have reported a diverse set of changes in learning and behavior. When trained to associate a tone with a foot shock, Rag2$^{-/-}$ mice show significantly less freezing when presented with the tone, indicating a fear learning deficit [59]. The freezing response was partially recovered in Rag2$^{-/-}$ that had been reconstituted with CD4+ T-cells. In a social interaction test, Rag2$^{-/-}$ mice spent significantly less time interacting with a conspecific mouse than wildtype mice, and this phenotype was also rescued with reconstitution of functional lymphocytes [60]. Reconstituted Rag2$^{-/-}$ also showed less anxiety behavior than naïve Rag2$^{-/-}$ mice, as measured by time in the open arm of an elevated plus maze. These studies demonstrate that lymphocytes can shape behavior outside of pathological conditions, though it remains unclear how this influence is exerted.

In this study we sought to determine whether the homeostatic transcriptome states of CNS glial cells require signals from lymphocytes. To that end, we acutely purified cortical oligodendrocytes, OPCs, astrocytes, and microglia by the immunopanning method from immunodeficient Rag2$^{-/-}$ mice and immunocompetent littermates. We performed RNA-sequencing to characterize the transcriptome profiles of each of the glial cell types. We found modest changes in gene expression among oligodendrocytes, though gross myelin development appears normal. However, we did identify altered localization of an RNA-binding protein, Quaking, in oligodendrocytes, which binds transcripts for a key myelin gene, *MBP* [61]. Microglia, OPCs, and astrocytes show little to no alterations in gene expression in the cortex, despite a previous study suggesting that lymphocyte depletion altered microglial gene signatures [62]. Overall, we find little evidence that lymphocytes influence CNS function by majorly altering the transcriptome profiles of microglia, astrocytes, and OPCs in the healthy cortex.

## Methods

### Experimental animals

All animal care and experimentation were approved by the Animal Research Committee at the University of California, Los Angeles (UCLA) under the approved protocol #R-16-080. We obtained Rag2$^{-/-}$ mice from Jackson Laboratory (B6.Cg-Rag2$^{tm1.1Cgn}$/J, #008449) and crossed with C57BL/6J to establish the breeding colony used in all sequencing and immunostaining experiments. We ordered 8-week-old male Rag2$^{-/-}$ mice (B6.Cg-Rag2$^{tm1.1Cgn}$/J, #008449) and controls (C57BL/6J, #000664) from Jackson Laboratory for western blot experiments. Mice were housed in autoclaved cages and received sterilized food and water. Both male and female mice were used for experimentation. We used 3-month-old mice for RNA-sequencing experiments, and approximately 1-year-old mice for RNAscope experiments.

### Purification of brain cells

Four classes of brain cells (microglia, OPCs, oligodendrocytes, and astrocytes) were purified using immunopanning, as described in Zhang 2016 and Zhang 2014 [63, 64]. Briefly, we anesthetized animals with isofluorane and performed transcardial perfusions with phosphate buffered saline (PBS) and subsequently dissected cortical grey matter. The tissue was digested enzymatically with papain (12 units/mL) at 34.5˚C for 45 minutes, followed by mechanical trituration to generate a single cell suspension. Cells were treated with enzymatic inhibitor to end

digestion. We incubated this single cell suspension for 10–15 minutes at room temperature on a series of petri dishes that were pre-coated with cell-specific antibodies. After incubation, we washed each dish with PBS to remove contaminants, applied TRIzol to release the RNA, and flash froze the resulting sample in liquid nitrogen for storage at -80˚C. We used the following series of antibodies to purify each cell class in this order: microglia, anti-CD45 x3 plates (BD Pharmingen 550539); OPCs, anti-PDGFRα x1 plate (BD Sciences 558774) (harvested for RNA-sequencing) and O4 hybridoma x2 plates (to further deplete OPCs, not harvested for RNA-sequencing); oligodendrocytes, GalC hybridoma x2-3 plates; astrocytes, HepaCAM x1 plate (R&D Systems MAB4108). Of note, anti-CD45 can also bind a population of peripheral macrophages, though this population was reduced by perfusion prior to brain dissection. The following sample sizes were collected for each cell class: astrocyte n = 12 [6 control, 6 Rag2--knockout (KO)], microglia n = 12 (6 control, 6 KO), OPC = 11 (6 control, 5 KO), and oligo-dendrocyte = 8 (4 control, 4 KO). Samples also included both males and females: astrocyte 7 male, 5 female; microglia 7 male, 5 female; OPC 7 male, 4 female; oligodendrocyte 5 male, 3 female.

## RNA-sequencing library construction and sequencing

RNA was purified from frozen samples using the miRNeasy kit (Qiagen 217004) according to the manufacturer's protocol. The resulting RNA was converted to cDNA and amplified using the Nugen Ovation RNAseq System V2 (Nugen 7102–32), and fragmented using a Covaris S220 focused-ultrasonicator (Covaris 500217). Final libraries were prepared using the NEB Next Ultra RNA Library Prep Kit (New England Biolabs E7530S) and NEBNext multiplex oli-gos for Illumina (NEB E7335S) according to manufacturer's protocol. Libraries from the same cell type from all mice were pooled and sequenced on the same lane using the Illumina Nova-Seq 600 System to obtain 23.2 ± 5.74 (s.d.) million 2x50 bp reads per sample. RNA integrity was measured using the 2200 TapeStation System (Agilent G2964AA) and the RNA high sensi-tivity assay (Agilent 5067–5579). All samples had RIN > 7, though some samples were out of the measurable range.

## Read alignment and quantification

We mapped the reads using the STAR package v2.7.8a and genome assembly GRCm39 (Ensembl, release 104) [65]. Samples had 73.7% ± 2.97 (s.d.) uniquely aligned reads. Reads were quantified using HTSeq v0.13.5 to obtain counts for downstream analysis [66]. Quanti-fied RNA-seq data can be found in S1 File.

## Differential gene expression analysis with DESeq2

We analyzed differential gene expression of each cell type using gene counts and the DESeq2 (v1.26.0) package in R [67]. We built our linear model using only two binary variables: sex and genotype. Full differential gene expression results are reported in the S2 and S3 Files. Male- and female-enriched genes in oligodendrocytes were also passed into the online database STRING to identify functional enrichment of these gene sets [68].

## Gene set enrichment analysis (GSEA)

We downloaded GSEA software from www.gsea-msigdb.org, version 4.2.3 [69, 70]. We used the default settings with the following exceptions. We entered normalized counts for our expression data, which we calculated using the DESeq2 functions estimateSizeFactors() and counts(). "Permutation type" was set to "gene_set". We built our own gene sets based on

scRNAseq analysis published in Pasciuto 2020 [62]. They sequenced cells from MHCII knockout mice which have a different form of lymphocyte deficiency [71]. We extracted the genes they found to be differentially expressed among all MHCII$^{-/-}$ microglia vs all control microglia. We made one gene set of upregulated genes and one gene set of downregulated genes. The gene sets were trimmed to the top 500 genes ranked by p-values to meet the recommended gene set size.

## Principal components analysis

Principal components analysis (PCA) was performed to visualize RNAseq results in a low-dimensional space. In R, we converted raw read counts using a log$_2$ transformation with the function "rlogcounts" followed by PCA using the function "prcomp". Resulting plots are shown in supplemental data, S2 Fig.

## RNAscope *in situ* hybridization

*In situ* hybridization of microglial markers was performed using the RNAscope Multiplex Fluorescent V2 Assay (ACDBio 323100). Brain tissue was harvested from approximately 1 year old mice (3 Rag2$^{+/+}$, 3 Rag2$^{-/-}$) after anesthetization with isoflurane and 10-minute transcardial perfusion with 4% paraformaldehyde. Brains were postfixed overnight in 4% PFA at 4°C, then dehydrated in 30% sucrose at 4°C until brains sank. Finally, brains were embedded in OCT compound (Fisher Scientific 23-730-571) and sectioned at 15 μm thickness and mounted onto Superfrost Plus slides (Fisher Scientific 12-550-15) before proceeding to the RNAscope assay. The assay was conducted as per the manufacturer's protocol. We used the following probes, formulated by ACD Bio: Mm-Tmem119 (cat. 472901), Mm-C1qa-C2 (cat. 441221-C2), and Mm-Junb-O1-C3 (cat. 584761-C3). The cerebral cortex was imaged with a 20x objective in both the upper cortex (layers 2–3) and lower cortex (layers 4–6) in both the motor and dorsal somatosensory regions. Images were quantified using ImageJ (2.0.0-rc-61/ 1.51n with Java 1.8.0_66) [72]. To quantify fluorescence intensity, regions of interest were manually drawn around microglial soma using microglia-specific markers *Tmem119* and *C1qa*, and intensity was measured using the "Measure" tool. For microglial-specific markers *Tmem119* and *C1qa*, we also quantified the area of staining by first applying an equal threshold to all images before measuring area with the "Measure" tool. Welch's t test was used to assess differences between *Rag2*$^{+/+}$ and *Rag2*$^{-/-}$ tissue.

## Data deposition

We deposited all gene expression data to the Gene Expression Omnibus, accession number GSE210580. To review the dataset, please go to https://www.ncbi.nlm.nih.gov/geo/query/acc. cgi?acc=GSE210580 and use token mtivisuenjgrxkf.

## Immunohistochemistry

Brain tissue was fixed and embedded as described for RNAscope. Brains were sectioned at 15–20 μm and directly mounted onto Superfrost Plus slides (Fisher Scientific 12-550-15). Sections were permeabilized with a blocking solution made of 0.2% Triton-X and 10% donkey serum in PBS at room temperature for 30 minutes, then they were rinsed and incubated with primary antibody diluted in blocking solution overnight at 4°C. The following day, sections were washed 3 times in PBS and incubated with secondary antibody at room temperature for 90 minutes, followed by 3 PBS washes. Finally, coverslips were added with a mounting solution containing DAPI. Primary antibodies: anti-MBP, 1:200 (Abcam ab7349), anti-APC clone CC1

(Millipore Sigma OP80); Secondary antibodies: anti-rat 647 (Invitrogen A48272), anti-rat 594 (Invitrogen A-21209), anti-mouse 488 (Invitrogen SA5-10166). Staining was quantified in 1-year-old mice (n = 3/group). MBP was quantified in coronal sections approaching the crossing of the anterior commissure (bregma 0 mm to 0.1 mm).

## Western blot

Whole-cell lysates from 2-month-old mouse cortex were lysed RIPA buffer (Thermo Fisher, cat #89901) containing EDTA-free protease inhibitor cocktail (Sigma, cat #4693159001), and centrifuged at 12,000 × g for 10 min to remove cell debris. Whole-cell lysates were then mixed with sodium dodecyl sulfate (SDS) sample buffer (Fisher, cat # AAJ60660AC) and 2-mercaptoethanol before boiling for 5 min. Samples were separated by SDS-polyacrylamide gel electrophoresis, followed by transfer to polyvinylidene difluoride membranes (Thermo Fisher, 88520) via wet transfer at 300 mA for 1.5 hours. Membranes were blocked with clear milk-blocking buffer (Fisher, cat #PI37587) for 1 hour at room temperature and incubated with primary antibodies against MBP (Abcam, cat #ab7349, dilution 1:1000), GAPDH (Sigma, cat #CB1001, dilution 1:5000), and PLP1 (Millipore, cat #MAB388, dilution 1:500) at 4˚C overnight. Membranes were washed with tris-buffered saline with Tween 20 (TBST) three times and incubated with either horseradish peroxidase-conjugated secondary antibodies (Mouse, Cell Signaling, cat #7076S; Rat, Cell Signaling, cat #7077S) (for MBP and PLP1) or Donkey anti-Mouse IgG (H+L) Highly Cross-Adsorbed Secondary Antibody, Alexa Fluor™ Plus 647 (Fisher, cat # PIA32787,1:1000) (for GAPDH) for 1 hour at room temperature. After three washes in the TBST buffer, SuperSignal™ West Femto Maximum Sensitivity Substrate (Fisher, cat #PI34095) was added to the membranes, and these signals were visualized using a ChemiDocTM MP Imaging system (BIO-RAD). Images were quantified in ImageJ using the plugin "bandandPeakQuantification" and normalized to Gapdh expression.

## Statistics

Differential gene expression and associated statistical testing was performed using DESeq2. Gene set enrichment analysis (GSEA) was performed using GSEA software v4.3.2, as described above. All other statistical comparisons were performed using Welch's t-test in Excel.

## Results

### RNA-sequencing of glia from Rag2$^{-/-}$ mice

*Rag2*$^{-/-}$ immunodeficient mice and wildtype immunocompetent control mice are typically housed in facilities with different levels of pathogen exposures and other environmental variables. To compare glial cell states in *Rag2*$^{-/-}$ and control mice with minimal environmental confounding factors, we established a single colony of *Rag2*$^{+/-}$ heterozygous mice housed in a clean facility for immunodeficient mice. We crossed heterozygous parents to generate *Rag2*$^{-/-}$ and *Rag2*$^{+/+}$ littermate pairs raised with the same maternal care in the same environment for RNA-sequencing (Fig 1A).

We purified each glial cell type from the mouse cerebral cortex using an immunopanning technique [63, 64]. Cells were separated into a single-cell suspension and then passed over plates coated with cell type specific antibodies that pull down the cell types of interest (Fig 1B). Compared with the traditional method of culturing glial cells in serum-containing media and separating them based on the layers in which each cell type is enriched, immunopanning allows acute purification of glial cells without exposure to serum and allows cells to remain much closer to a physiological state [63, 64]. Using this method, we collected microglia (anti-

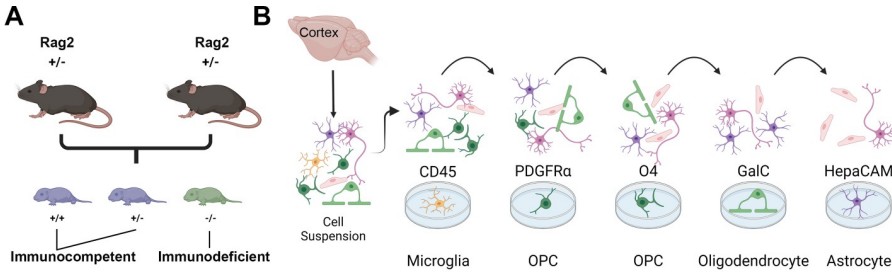

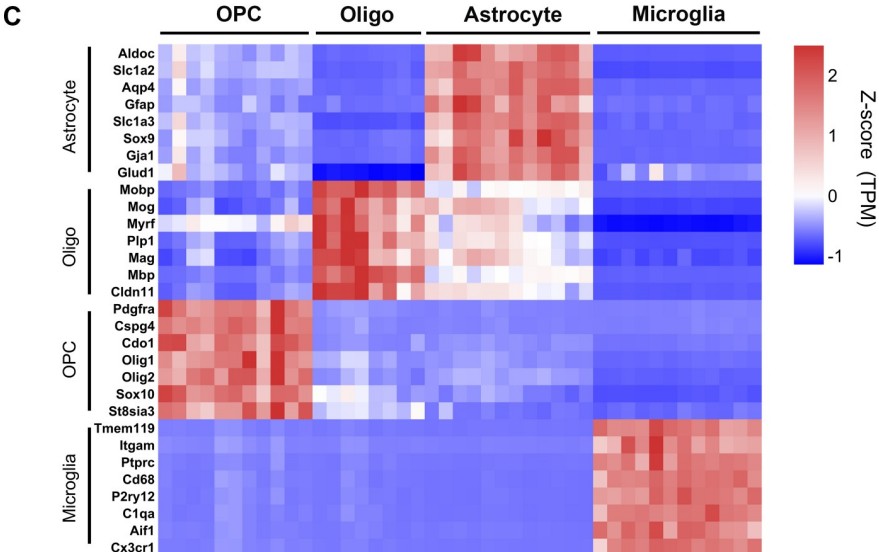

**Fig 1. Acute purification of glial cell populations from immunodeficient mice.** A) Breeding schematic; Rag2$^{+/-}$ parents bore offspring that were immunocompetent (Rag2$^{+/+}$ and Rag2$^{+/-}$) or immunodeficient (Rag2$^{-/-}$). All immunodeficient mice and littermate controls were maintained in the same environment. B) Immunopanning schematic; a single-cell suspension was generated from the cerebral cortex, then passed over a series of plates coated with cell type specific antibodies to enrich for specific glial cell populations. C) Heatmap showing enrichment of cell-specific markers (rows) among the glial samples that we harvested and sequenced (columns); gene expression is quantified as transcripts per million (TPM), and each gene was further normalized to a z-score, defined as (expression in the sample—the average expression across all samples)/standard deviation; *i.e.*, the number of standard deviations from the average.

CD45), astrocytes (anti-HepaCAM), oligodendrocytes (GalC hybridoma), and OPCs (anti-PDGFRA) from 4–6 littermate pairs of *Rag2*$^{-/-}$ and *Rag2*$^{+/+}$ mice and performed RNA-sequencing. Using a panel of cell type-enriched genes, we found that glial samples enriched via immunopanning have low levels of contamination from other cell types (Fig 1C).

## Lymphocyte deficiency affects the oligodendrocyte transcriptome and Quaking RNA-binding protein localization

We analyzed each cell type for differential gene expression using DESeq2. We found five or fewer differentially expressed genes (multiple comparison adjusted p-value <0.05) in astrocytes, OPCs, and microglia from immunocompetent vs. immunodeficient *Rag2*$^{-/-}$ mice. Of note, the gene *Iftap* (encoding intraflagellar transport associated protein) overlaps with *Rag2* in the genome, and *Iftap* is significantly downregulated in all four cell types analyzed. This observation suggests that the coding and/or regulatory sequences of *Iftap* is disrupted in *Rag2*$^{-/-}$ mice.

In contrast to the other three cell classes, oligodendrocytes showed more differentially expressed genes: 19 upregulated and 180 downregulated genes, of which 16 upregulated genes and 71 downregulated genes are protein-coding (Fig 2A). This suggests a role of peripheral lymphocytes in maintaining some aspect of oligodendrocyte molecular signatures.

Among the downregulated genes is *Man1a2*, a gene encoding an enzyme involved in N-glycosylation of peptides. N-glycosylation occurs on many important peptides expressed by oligodendrocytes, including myelin oligodendrocyte glycoprotein (MOG) [73]. $Rag2^{-/-}$ oligodendrocytes also downregulate *Spx*, which encodes the neuropeptide spexin, also known as neuropeptide Q. Spexin has been implicated in a variety of functions, including nociception and feeding behaviors, though its role in oligodendrocytes has not been described [74].

The upregulated genes include *Sema3b*, a member of the semaphorin family of genes that encode axon guidance cues. Interestingly, we also see observed differential expression of *Ppia*, which encodes an enzyme that catalyzes isomerization of peptide bonds. *Ppia* is sometimes referred to as a housekeeping gene and used as a reference gene in real time quantitative PCR, so its differential expression is interesting to note [75].

Upon further inspection of oligodendrocytes in the immunodeficient mice at the protein level, we found a striking change in CC1, a canonical marker of mature oligodendrocytes. We find that $Rag2^{-/-}$ show altered cellular distribution of CC1 within white matter (Fig 2B). In control mice, CC1 labels oligodendrocyte soma, as well as a number of processes. Somatic expression of CC1 remains in $Rag2^{-/-}$ mice, but CC1+ processes virtually disappear. CC1 antibodies specifically recognize the RNA-binding protein Quaking (QKI), specifically isoform 7 [76]. QKI in oligodendrocytes is known to bind myelin basic protein (*MBP*) mRNA, and QKI disruption was previously shown to prevent *MBP* export to cytoplasmic processes, ultimately altering myelination [61].

To assess whether MBP levels were altered in conjunction with QKI localization, we performed immunohistochemistry experiments and measured MBP in three myelin-rich regions: the corpus callosum (p = 0.44, mean[control, KO] = 922.6, 762.4, SD[control, KO] = 288.1, 46.3), anterior commissure (p = 0.46, mean[control, KO] = 809.7, 762.0, SD[control, KO] = 80.1, 61.6), and striatum (p = 0.32, mean[control, KO] = 729.7, 698.6, SD[control, KO] = 41.2, 10.1). We found no difference between immunocompetent and immunocompromised mice in any of these regions (Fig 3A). To assess myelin protein levels using another method, we used western blots. Again, we found no difference in expression in MBP (p = 0.69, mean[control, KO] = 4.25, 4.38, SD[control, KO] = 0.55, 0.33) or another key myelin protein, myelin proteolipid protein, PLP (p = 0.61, mean[control, KO] = 0.20, 0.19, SD[control, KO] = 0.018, 0.021; Fig 3B). This suggests that lymphocytes are not required for gross myelination.

## Sexual dimorphism in glial gene expression

Given the increased incidence of autoimmune disorders in women compared to men, we also examined differential gene expression associated with sex in our dataset. We found genes that were significantly associated with sex in each cell type, many of which were located on the X or Y chromosome (e.g. *Kdm5d*, *Uty*, *Eif2s3y*). Interestingly, oligodendrocytes again showed the most robust difference with 143 female-enriched genes, and 136 male-enriched genes (S3 File). Using a database of protein-protein interactions, STRING, we found that male-enriched genes showed functional enrichment for SNAP/SNARE and endosome terms, while female-enriched genes had functional enrichment for voltage-gated channel and neuronal system terms [68]. Among the other cell types we found the following numbers of female-enriched/male-enriched protein-coding genes: astrocytes 1/13, microglia 3/4, OPCs 2/6. We observed

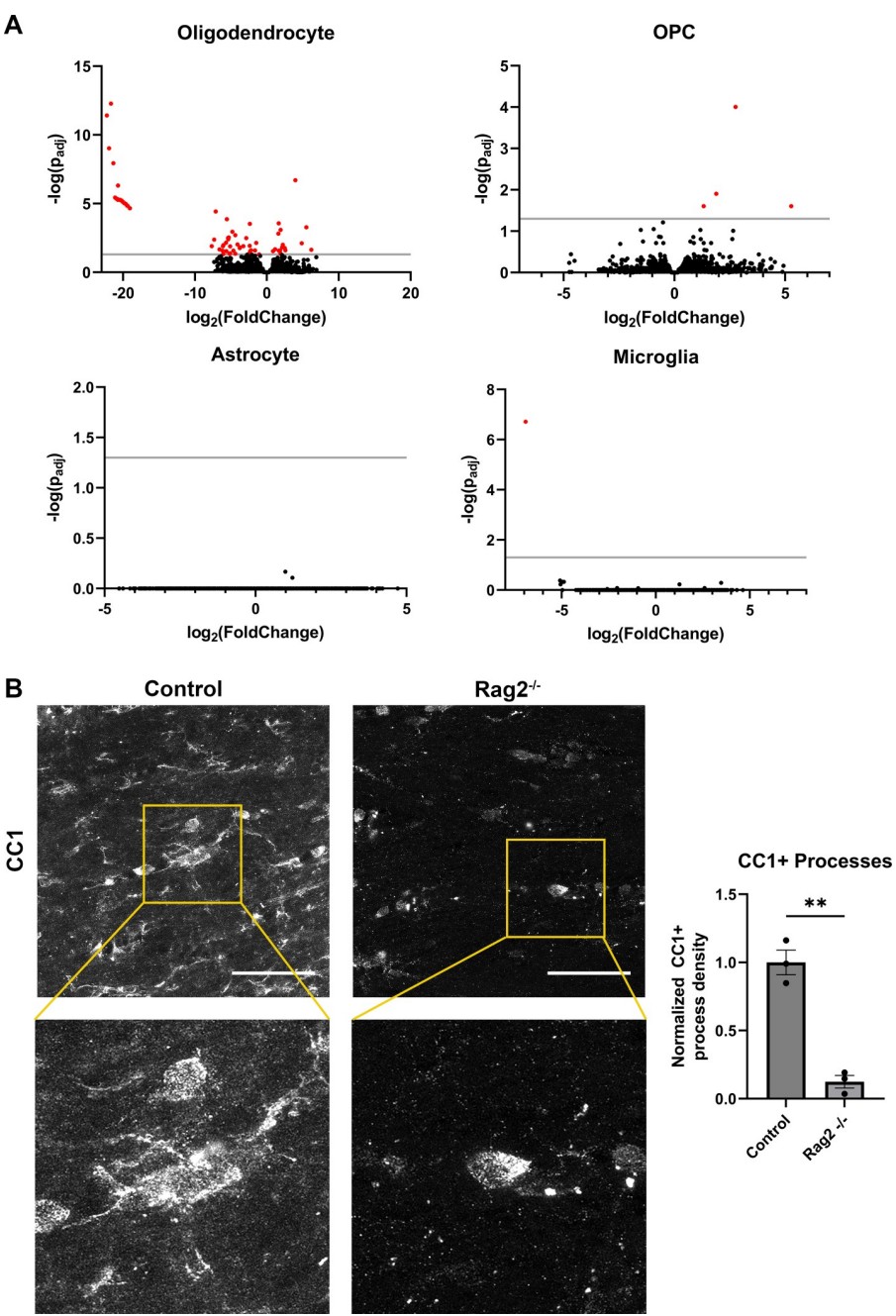

**Fig 2. Differential gene expression of glial cells in Rag2$^{-/-}$ mice.** A) Volcano plots displaying differential gene expression analysis of oligodendrocytes (top-left), OPCs (top-right), astrocytes (bottom-left), and microglia (bottom-right). Differential gene expression was analyzed using DESeq2, and the resulting fold change and statistical significance are plotted on the x and y axes respectively. Red: p < 0.05; gray line: p = .05. B) Loss of CC1 expression in cellular processes. Top: Representative images of CC1 in the corpus callosum of control (left) and Rag2$^{-/-}$ (right) mice; Bottom: Insets of control showing CC1+ processes extending from a CC1+ cell bodies and KO showing only CC1 + soma. Right: Quantification of CC1+ process density, normalized to average control levels (right, p = 0.0034; mean [control, KO] = 1.0, 0.1246; SD[control, KO] = 0.157, 0.081; error bars = SEM). Scale bar = 50 μm.

that *Kdm6a* expression was significantly higher in female astrocytes compared to males, which agrees with our previous findings from RNAseq of human astrocytes [77].

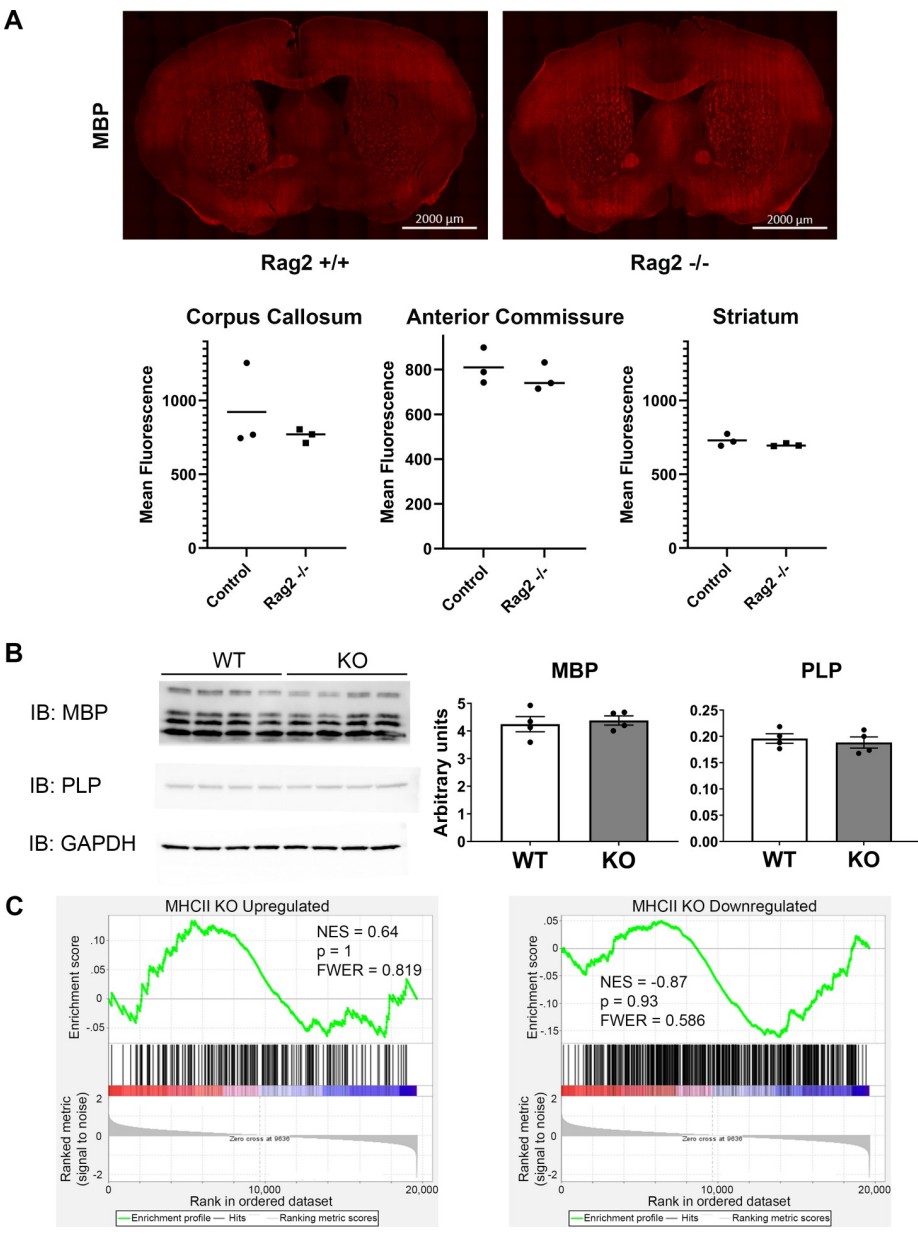

**Fig 3. Normal myelin proteins and microglial gene signature in Rag2$^{-/-}$ mice.** A) Immunostaining of MBP. Top: representative images of MBP immunofluorescence in control (left) and Rag2$^{-/-}$ (right) mice. Bottom: quantification of MBP fluorescence in 3 myelin rich regions; no significant differences. B) Western blots of key myelin proteins MBP and PLP. Left: Images of Western blot of MBP (top), PLP (middle), and reference protein GAPDH (bottom), Right: Quantification of signal intensity for MBP and PLP, normalized to GAPDH signal. No significant differences. All error bars = SEM. C) GSEA output of genes upregulated (left) or downregulated (right) in MHCII KO microglia reported in Pasciuto 2020. Neither MHCII KO up- nor downregulated genes are significantly enriched in a comparison of Rag2$^{+/+}$ *vs.* Rag2$^{-/-}$ microglia (p = 1, 0.93); NES = normalized enrichment score, FWER = family wise error rate.

## Lymphocyte deficiency does not affect the maturation of microglia

Our RNA-sequencing data revealed that the expression of mature microglia markers such as *Cx3cr1*, *Tmem119*, *P2ry12*, and *Aif1* do not significantly differ between immunocompetent vs. immunodeficient Rag2$^{-/-}$ mice. This is somewhat surprising given microglia are the brain resident immune cells, and they express high levels of receptors for immune signaling molecules

that peripheral lymphocytes could use to pass signals into the brain. To further assess markers of microglia in immune-compromised mice, we performed RNA and protein level analysis of microglial markers. First, we reanalyzed our RNAseq data by identifying a panel of microglial genes, and we generated an aggregate expression score for each sample (Panel C in S1 Fig). There were no differences between Rag2$^{-/-}$ and control mice. Second, we performed RNAscope *in situ* hybridization to visualize the expression of genes found in mature microglia: *Tmem119*, *C1qa*, and *Junb*. Once again, we found no differences in gene expression (Panels A, B in S1 Fig). Lastly, we performed immunohistochemistry to assess protein levels of the microglial markers Iba1, P2ry12, and Cd68, and we continued to detect no differences in the Rag2$^{-/-}$ mice (Panel D in S1 Fig).

Given the striking lack of aberration among these brain resident immune cells, we asked whether microglia responded to other changes in peripheral immunity. One such animal model knocks out a set of major histocompatibility complex class II (MHCII) genes, which results in the loss of CD4+ T cells [71]. Rag2$^{-/-}$ mice, in contrast, lack all mature lymphocytes, including all T cells and B cells for a more complete depletion of adaptive immune cells. MHCII$^{-/-}$ animals show substantial differences in microglia transcription, including downregulation of highly expressed microglial genes including *P2ry12*, *Itgb5*, and *Tgfb1* [62]. To make a direct comparison between microglial gene expression in total lymphocyte-deficient *Rag2$^{-/-}$* mice and CD4+ T lymphocyte-deficient MHCII-knockout mice in Pasciuto 2020, we took a more systematic approach to assess whether the microglial differential gene expression signature observed in T lymphocyte-deficient MHCII-knockout mice is enriched in microglia from *Rag2$^{-/-}$* mice. We took all the differentially expressed genes in microglia from MHCII-knockout mice from the Pasciuto study and performed gene set enrichment analysis (GSEA) using our RNA-seq data [62]. We found that the up- and down-regulated genes identified in their study did not show global enrichment in our dataset (Fig 2C). That is to say, upregulated genes in MHCII-knockout microglia did not trend toward higher expression in *Rag2$^{-/-}$* microglia, nor did downregulated genes in MHCII-knockout microglia trend toward higher expression in immunocompetent microglia in this current study. This contrast suggests that the exact complement of peripheral lymphocytes can exert highly varied and perhaps surprising changes in the brain.

## Discussion

We generated transcriptomic data of acutely purified glial cells from mice lacking adaptive immune cells and their immunocompetent littermates. We found differentially expressed genes among oligodendrocytes, while the transcriptome of microglia, astrocytes, and OPCs remained largely unaltered by the lack of lymphocytes. In oligodendrocytes, we found altered localization of the RNA-binding protein QKI. Given previous reports of microglia changes in immune-compromised mice [62], we validated our sequencing results with *in situ* hybridization and immunohistochemistry of microglial markers and found no differences in *Rag2$^{-/-}$* mice. We also performed a bioinformatic analysis of the microglia that failed to detect the previously reported gene signature found in a different T lymphocyte deficiency model. Together, these data shed light on the impacts of peripheral immune cells on the brain, and suggest underappreciated interactions between oligodendrocytes and lymphocytes.

### Molecular profile of oligodendrocytes in the absence of adaptive immunity

Oligodendrocytes were the only cell class in this study to show appreciable differential expression in *Rag2$^{-/-}$* mice. The differentially expressed genes have a wide variety of functional roles in the brain that defy easy classification. Pathway analysis of gene expression, including gene

ontology and gene set enrichment analysis, failed to identify larger patterns among these genes. Among the differentially expressed genes were axon guidance cues (*Sema3b*), glycosylation enzymes (*Man1a2*), neuropeptides (*Spx*), transcription factors (*Gli1*), and proteasome components (*Psmd5*).

At the protein level, we found differences in the expression of RNA-binding protein Quaking isoform 7, as shown with the classical oligodendrocyte marker CC1. QKI is important for trafficking various mRNAs, including the key myelin protein gene *MBP*. In Rag2$^{-/-}$ white matter, QKI7 no longer enters the processes, which may be relevant for delivering important oligodendrocyte transcripts like *MBP* to sites of myelination. We find that the overall levels of MBP do not change in adult Rag2$^{-/-}$ mice, but future studies could investigate potential changes during myelination in development that may underlie the behavioral phenotypes observed in these mice.

The link between peripheral immune state and oligodendrocyte transcription may provide a fruitful new avenue for understanding their interactions under pathological conditions. Lymphocyte interactions with oligodendrocytes and their myelin sheaths have long been suspected to be central to the demyelinating pathology of multiple sclerosis [51]. To our knowledge, this is the first evidence that oligodendrocytes are affected by lymphocytes in the healthy brain. Further elucidation of the interactions between oligodendrocytes and lymphocytes in homeostatic conditions could improve our understanding of how these interactions become maladaptive in a disease state.

## Homeostatic microglia are unaltered in total lymphocyte deficiency

In this study, we find that microglia in adult *Rag2*$^{-/-}$ mice under homeostatic conditions are indistinguishable from microglia in immunocompetent mice in their transcriptome profiles. This finding comes in surprising contrast to a previously published report that CD4+ T-cells were required for microglial maturation [62]. In that study, investigators used an MHCII knockout mouse model that lacks several genes that make protein products for the major histocompatibility complex 2. MHCII$^{-/-}$ mice specifically lack CD4+ T-cells, while maintaining other lymphocyte populations including CD8+ T-cells and B cells [71]. Single-cell sequencing of these cells found that MHCII$^{-/-}$ microglia downregulated highly expressed microglial markers including *P2ry12*, *Itgb5*, and *Tgfb1*. They therefore conclude that microglia from MHCII$^{-/-}$ mice are arrested in an immature state.

In contrast, the *Rag2*$^{-/-}$ model used in the current study results in the loss of all mature lymphocytes, including CD4+ T cells, CD8+ T cells, and B cells [58]. Despite a more comprehensive loss of adaptive immune cells, microglia from these mice did not show major transcriptional perturbations. The divergence in these two immunodeficiency models poses several interesting possibilities that should be explored in future studies. First, various lymphocyte classes may exert different or even opposing influences on brain cells. Perhaps the MHCII$^{-/-}$ microglia are altered not just by the absence of CD4+ T-cells, but also the influence of remaining T-cells and B-cells that would otherwise face regulation by CD4+ T-cells. This model could be compatible with unperturbed Rag2$^{-/-}$ microglia, where the relative balance of lymphocyte classes is maintained (i.e. all present or all absent). Alternatively, MHCII$^{-/-}$ may directly alter microglia. Microglia can express MHCII genes and become antigen presenting cells, whereas Rag2 is a lymphocyte-specific protein. However, microglial MHCII expression is largely thought to occur in pathological conditions, and little if any MHCII protein expression has been found in homeostatic microglia. Furthermore, Pasciuto et al. show that reintroduction of CD4+ T-cells to MHCII$^{-/-}$ slice culture can partially rescue some downregulated microglia genes, which argues for a causal role of lymphocytes in MHCII$^{-/-}$ microglia. Still, loss of

MHCII genes may exert direct effects on microglia that are absent in the Rag2$^{-/-}$ model. The distinctions between the MHCII and Rag2 models serve as a fruitful ground for further dissection of lymphocytic influence in the brain.

Neuro-immune interactions represent an exciting frontier of neurobiology that was previously overlooked. While modern studies now suggest influential roles of peripheral immune cells in brain function and behavior [4, 10, 59, 60, 78], it is important to understand the extent and the limits of this influence. These data provide important insight into which brain cells might interface with adaptive immune cells in non-pathological conditions. Of equal importance, this study also points to limits of adaptive immune influence in the central nervous system, and it insinuates that various peripheral immune cells may wield distinct influences within the central nervous system that remain to be explored.

## Supporting information

**S1 File. Rag2 glia gene expression.** Gene expression of glia (astrocytes, microglia, oligodendrocytes, and OPCs) from Rag2$^{-/-}$ mice and immunocompetent controls, quantified by transcripts per million (TPM).
(XLSX)

**S2 File. Rag2 glia differential gene expression by genotype.** Differential gene expression analysis results comparing glia (astrocytes, microglia, oligodendrocytes, and OPCs) from Rag2$^{-/-}$ mice and immunocompetent controls using DESeq2.
(XLSX)

**S3 File. Rag2 glia differential gene expression by sex.** Differential gene expression analysis results comparing glia (astrocytes, microglia, oligodendrocytes, and OPCs) from females vs. males.
(XLSX)

**S1 Fig. Normal microglial markers at the RNA and protein levels.** A) Example RNAscope images of microglial markers Tmem119 (green), C1qa (orange), and Junb (red) and composites including DAPI (blue) in the cerebral cortex. Top: Rag2$^{+/+}$ immunocompetent; bottom: Rag2$^{-/-}$ immunodeficient. Scale bar = 100 μm. B) Quantification of RNAscope based on fluorescence intensity (top row) or area (bottom row) of microglial genes Tmem119 (Mean fluorescence: p = 0.79, mean[control, KO] = 10.6, 10.0, SD[control, KO] = 0.54, 3.34; Area: p = 0.68, mean[control, KO] = 74.9%, 66.0%, SD[control, KO] = 18.3, 28.6), C1qa (Mean fluorescence: p = 0.43, mean[control, KO] = 63.9, 72.0, SD[control, KO] = 8.0, 13.5; Area: p = 0.77, mean[control, KO] = 87.7%, 83.9%, SD[control, KO] = 15.2, 14.4), and Junb (Mean fluorescence: p = 0.63, mean[control, KO] = 5.67, 4.56, SD[control, KO] = 3.37, 0.74) from Rag2$^{-/-}$ and control mice (n = 3 KO, 3 WT). Error bars = SEM. C) Expression of microglia marker genes. Left: Heatmap of expression of 12 microglial marker genes, shown as transcripts per million (TPM) with z-score normalization across all samples, defined as (expression in the sample—the average expression across all samples)/standard deviation. Right: Microglial maturation score quantification, defined as the average z-score across all genes for each sample (p = 0.81, mean[control, KO] = -0.065, 0.065, SD[control, KO] = 1.03, 0.35). D) Immunostaining of microglial proteins. Top: representative images of Iba1, P2ry12, and Cd68 from Rag2$^{+/+}$ and Rag2$^{-/-}$ mice. Scale bar = 50 μm. Bottom: Quantification of fluorescence intensity for Iba1 (p = 0.67, mean[control, KO] = 712, 743, SD[control, KO] = 15.8, 122.4), P2ry12 (p = 0.57, mean[control, KO] = 466, 506, SD[control, KO] = 77.1, 80.6), and Cd68 (p = 0.89, mean[control, KO] = 373, 363, SD[control, KO] = 17.6, 99.1). Error bars = SEM.
(TIF)

**S2 Fig. PCA plots of Rag2 glia.** PCA of RNA sequencing data from glia from Rag2$^{-/-}$ and control mice. Red = female, blue = male.
(TIF)

## Acknowledgments

We thank Michael Sofroniew, Baljit Khakh, Michael Gandal, and Jessica Rexach for advice. We thank the Eli and Edythe Broad Center of Regenerative Medicine and Stem Cell Research, UCLA BioSequencing Core Facility for their services.

## Author Contributions

**Conceptualization:** Mitchell C. Krawczyk, Ye Zhang.

**Formal analysis:** Mitchell C. Krawczyk, Lin Pan.

**Funding acquisition:** Mitchell C. Krawczyk, Ye Zhang.

**Investigation:** Mitchell C. Krawczyk, Lin Pan, Alice J. Zhang.

**Project administration:** Ye Zhang.

**Resources:** Ye Zhang.

**Supervision:** Ye Zhang.

**Visualization:** Mitchell C. Krawczyk, Lin Pan.

**Writing – original draft:** Mitchell C. Krawczyk, Ye Zhang.

**Writing – review & editing:** Mitchell C. Krawczyk, Ye Zhang.

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
