## [Decision Letter · Decision Letter 0]

3 Oct 2022

PONE-D-22-23915Lymphocyte deficiency alters the transcriptomes of oligodendrocytes, but not astrocytes or microgliaPLOS ONE

Dear Dr. Zhang,

Thank you for submitting your manuscript to PLOS ONE. After careful consideration, we feel that it has merit but does not fully meet PLOS ONE’s publication criteria as it currently stands. Therefore, we invite you to submit a revised version of the manuscript that addresses the points raised during the review process.

As you can see from both reviewers' comments, there are concerns about the statistics and analysis of the results, and the relationship between the Rag2 and MHCII KOs, and importantly the lack of description of the most significant changes mentioned in the manuscript, regarding the oligodendrocytes.

We look forward to receiving your revised manuscript.

Kind regards,

Stella E. Tsirka

Academic Editor

PLOS ONE

Journal Requirements:

"I have read the journal's policy and the authors of this manuscript have the following competing interests: Ye Zhang consulted for Ono Pharmaceutical."

"This work is supported by the Achievement Rewards for College Scientists Foundation Los Angeles Founder Chapter and the National Institute of Mental Health of the National Institutes of Health (NIH) Award T32MH073526 to M.C.K, the National Institute of Neurological Disorders and Stroke of the National Institute of Health (NIH) R00NS089780, R01NS109025, the National Institute of Aging of the NIH R03AG065772, the National Institute of Child Health and Human Development P50HD103557, National Center for Advancing Translational Science UCLA CTSI Grant UL1TR001881, the W. M. Keck Foundation Junior Faculty Award, UCLA Eli and Edythe Broad Center of Regenerative Medicine and Stem Cell Research (BSCRC) Innovation Award, the UCLA Jonsson Comprehensive Cancer Center and BSCRC Ablon Scholars Program, and the Friends of the Semel Institute for Neuroscience & Human Behavior Friends Scholar Award to Y. Z"

"This work is supported by the Achievement Rewards for College Scientists Foundation Los Angeles Founder Chapter and the National Institute of Mental Health of the National Institutes of Health (NIH) Award T32MH073526 to M.C.K, the National Institute of Neurological Disorders and Stroke of the National Institute of Health (NIH) R00NS089780, R01NS109025, the National Institute of Aging of the NIH R03AG065772, the National Institute of Child Health and Human Development P50HD103557, National Center for Advancing Translational Science UCLA CTSI Grant UL1TR001881, the W. M. Keck Foundation Junior Faculty Award, UCLA Eli and Edythe Broad Center of Regenerative Medicine and Stem Cell Research (BSCRC) Innovation Award, the UCLA Jonsson Comprehensive Cancer Center and BSCRC Ablon Scholars Program, and the Friends of the Semel Institute for Neuroscience & Human Behavior Friends Scholar Award to Y. Z. The funders had no role in study design, data collection and analysis, decision to publish, or preparation of the manuscript."

Reviewers' comments:

Reviewer's Responses to Questions

**Comments to the Author**

1. Is the manuscript technically sound, and do the data support the conclusions?

Reviewer #1: Yes

Reviewer #2: Partly

2. Has the statistical analysis been performed appropriately and rigorously? 

Reviewer #1: Yes

Reviewer #2: No

3. Have the authors made all data underlying the findings in their manuscript fully available?

Reviewer #1: Yes

Reviewer #2: No

4. Is the manuscript presented in an intelligible fashion and written in standard English?

Reviewer #1: Yes

Reviewer #2: Yes

5. Review Comments to the Author

Reviewer #1: The authors set out to investigate whether in the healthy brain, lymphocytes play a role in the transcriptomic phenotype of glial cells. The study has solid foundation in the idea that adaptive immune changes correlate with changes to glial cells in various disease states. To answer the question the authors use lymphocyte (Rag2-/-) deficient mice in naïve states as well as control mice and conduct RNA sequencing on acutely purified microglia, astrocytes, OPCs, and oligodendrocytes. The sequencing reveals that once lymphocytes are eliminated there are no transcriptomic differences in glia except for modest differences seen in the oligodendrocytes, though gross morphology of WM remains the same. Although there is space and need for a publication of this kind, as it stands the publication is lacking clarity as to the overall impact of the findings and the authors must expand on the topic.

Reviewer #2: In this study, the authors sought to investigate whether lymphocytes are regulating the molecular phenotypes of glial cells (OPCs, oligodendrocytes, microglia, and astrocytes) in the healthy brain. They used a Rag2-/- mouse model (which results to the loss of all mature lymphocytes, including CD4+ T cells, CD8+ T cells, and B cells) to characterize the molecular phenotype of glial cells through RNA sequencing. The authors reported no changes in the molecular signatures of the majority of the glial cells, with the exception of oligodendrocytes. Considering the emerging interest on neuroimmune interactions (either innate or adaptive immune system) and the critical role they have upon the brain homeostasis and behavior, this study will help to further delve into potential connections between lymphocytes and glial cells in the CNS.

However, there are some major concerns as well as minor suggestions about the study:

1. The authors in the introduction section (and specifically lines 44-52) focus on studies related to meningeal lymphocyte roles in the brain. It is understandable that they should be distinguished from the peripherally recruited lymphocytes, however there is no other reference of the meningeal lymphocytes again in the manuscript and no distinction of them in the experiments they performed. Is there a specific reason the authors are underscoring this lymphocytic population? If not, then I would recommend using this space for the following suggestions (see end of comment 1).

Moreover, in the next paragraph (lines 56-72), the authors provide a general background about functions of glial cells in health and disease. I agree that it is important to introduce the glial cells, since it is an important aspect of this study, however It should not be narrated as a glial section from a review paper. My recommendation would be to focus on the neuroimmune interactions and the effects they have on glial cells and brain homeostasis, and importantly provide more information about the roles of lymphocytes in CNS health and disease-since this is the central scope of the paper (should be significantly expanded more than just the 81-83 lines). Along the same lines the authors should provide more information about the Rag2-/- animal model in the introductory section, and specifically discuss previous findings regarding the cellular, molecular and behavioral mechanisms shown in the literature (i.e. refs 61-62). The reader should be able to understand the current gap in knowledge and how this paper seeks to contribute on that.

2. The authors mention in lines 92-94: «Overall, we find little evidence that lymphocytes influence CNS function by majorly contributing to the cellular states of glial cell types in the healthy brain». I find this inaccurate for 3 reasons: i) this study focuses only in tissue isolated from cerebral cortex (excluding the remaining brain areas), ii) the RNA sequencing analysis portrays molecular signatures and not cellular states of glial cells, and iii) the molecular changes found in the oligodendrocyte populations should have been further characterized to be able to conclude on that (as later will be discussed).

3. In the materials & methods section (line 116), the anti-CD45 also targets a small population of resident macrophages (~1-3%). This should be depicted in the manuscript.

4. With regard to Figure 1 data the authors:

- Have not included information regarding either in figure legend or in the methods what z-score depicts and based on which control group the heatmap scale was made (increase or reduction of TPM compared to what control).

- I would recommend including more marker genes (at least 4 more classic markers) for each glial cluster.

- I would recommend the authors to provide a dot-plot analysis displaying the average expression levels (Change to avg. exp. scale), as well as the percentage of cells within each cell cluster expressing each marker gene (% Expression), split in groups of Rag2-/- vs Rag2+/+ mice. This information is needed to grasp the molecular signature of each group as well as whether the % of cells expressing the markers changes.

5. As in Figure 1, in Figure 2 more information about the volcano-plots should be included in the figure legend and the material & methods section.

6. The authors demonstrate an important finding depicting that lymphocyte deficiency dramatically affects the oligodendrocyte transcriptome. However, instead of focusing on this novel finding and try to strengthen their hypothesis with supplementary experiments, they only use one and a half panel [2A (Oligodendrocytes) and 2B] and they conclude in lines 244-245 that: «Despite the change in gene expression, gross patterns of myelin appear unchanged in Rag2-/- mice based on immunofluorescence of myelin basic protein (Fig 2B)».

Therefore, there are some major points that the authors should improve for this section:

i) Perform RNA expression analysis (i.e. RT-qPCR) to validate some things that showed up in their RNA sequencing analysis.

ii) Then the next step would be to check protein expression (i.e. immunoblot analysis of some classic myelin or generally mature oligodendrocyte markers).

iii) The authors provide 2 representative immunofluorescent staining for MBP, which brings up several issues:

- There are no graphs or statistics to support their claims

- These images actually depict a decrease in Rag2-/- mice

- But even if the representative images are wrongly selected, there is no information which area of the cortex is this. The white matter can be dramatically different depending on the area and the bregma coordinates these sections are from.

- The reader should be able to appreciate a larger cortical area (use lower magnification image and include insets with higher magnification)

- I would strongly recommend quantification, if there has not been performed already, which should be normalized per area (or use integrated analysis).

7. As mentioned in comment 6, the authors decided to not follow the innovative results they had from the RNA sequencing (Figs 1 and 2), but instead they used the whole Figure 3 to compare their findings on microglia with the previous publication of Pasciuto et al., Cell (2020). In my opinion the purpose and the flow of experiments in a study should not be determined by another study but on the hypothesis the authors have. On many occasions (lines 254-259) in the results section the authors compared their results with the previous study (it is more usual to do so in the discussion section), feeling as the sole purpose of this study was to prove the Pasciuto publication wrong. I would have been a lot more supportive on the narrative that the authors decided to take in Figure 3, if there was conclusive data of no effects of Lymphocytes upon the Oligodendrocytes (and as a consequence the authors sought investigate the microglia in more depth). However, the authors performed only a superficial characterization of Oligodendrocytes, and therefore decided to neglect the novel findings of their RNAseq study, making hard to follow the exact hypothesis of the study.

8. Regarding Figure 3, as previously mentioned in comment 6, the authors should also include either in the figure legend or the materials section, the following information:

- How the quantification of RNAscope analysis was performed.

- Which area of the cortex was analyzed

- It should also be depicted on the y axis of the graphs that the mean fluorescent intensity was normalized to the area.

9. The authors in the first sentence of the discussion (lines 313-314): In this study, we find that microglia in adult Rag2-/- mice under homeostatic conditions are indistinguishable from microglia in immunocompetent mice. However, apart from the RNA sequencing data and the RNA expression of Tmem119 and C1qa, there is no other support for the “indistinguishable” phenotype the authors claim. I would recommend characterization of supplementary microglial markers not only on a RNA level (some classic markers for RT-qPCR: P2RY12, PTPRC, CX3CR1, CTSS, LPAR6, CD68, ARHGAP24, ITGAM, AIF1), but most importantly for protein expression experiments. In my point of view, what the authors depict on this study is that there is only an indication of no substantial molecular signature differences in microglia during lymphocyte deficiency, which remains to be further examined with the aforementioned experiments (especially when comparing these results with the Pasciuto et al. publication, which has dedicated a large palette of experimental approaches to conclude to their findings).

10. Furthermore, in Fig.3a-b based on the RNAscope in situ hybridization in Rag2-/- and Rag2+/+ brains, the authors conclude that the maturation of microglia is unaffected by the lymphocyte deficiency. However, again this cannot be concluded just by fluorescent quantification of just three RNA expression markers, and no protein analysis. I would at least request the authors to utilize their RNAseq expression dataset to complement these findings with a comparative transcriptional analysis using a wide range of microglial maturation genes from Rag2-/- vs the Rag2+/+ mice, in order to get a general score.

11. Of importance, a clarification is required for the Fig.3c. The authors took all the differentially expressed genes in microglia from the MHCII-knockout mice from the Pasciuto study and performed gene set enrichment analysis (GSEA) using their RNA-seq data. Based on that the authors found that the up- and down-regulated genes identified in this study did not show global enrichment in this dataset. The question is, was the whole cluster of microglia used for this GSEA analysis or just the microglial subcluster 3 that is mentioned in the Pasciuto study?

12. In the discussion the authors suggest possible explanations of the different results shown on this study compared to the Pasciuto et al., publication. To this end, it would be important to further describe the differences between the Rag2 KO mice and the MHCII KO used on the other study. This way the reader would be able to appreciate the findings of this study and the deviations between the studies.

13. The authors should include a statistics section in the methods. Also, summary statistics, the data points behind means, medians and variance measures should be available.

6. PLOS authors have the option to publish the peer review history of their article (what does this mean?). If published, this will include your full peer review and any attached files.

Reviewer #1: No

Reviewer #2: **Yes: **A.G. Kokkosis

---

## [Author Response · Author response to Decision Letter 0]

21 Nov 2022

Response to Review: 

We would like to express our gratitude for the fair, thoughtful, and critical feedback from the reviewers that greatly improved our manuscript. After carefully reading their comments, we made the following changes to our manuscript. First, we will highlight the major new findings, followed by a point-by-point response to each review. 

Major Revisions

1. We found a protein level change in Rag2-/- oligodendrocytes, which coincides with the transcriptomic differences we find in this cell class. Oligodendrocyte marker CC1 is detected in oligodendrocyte soma as well as processes in wildtype mice, but they are restricted to the soma in Rag2-/- mice. CC1 labels the RNA-binding protein Quaking[1], which regulates the transport of transcripts for the key myelination gene MBP. Page 13-14, Fig 2B.

2. We deepened our characterization of Rag2-/- microglia with immunostaining of microglial proteins P2ry12, Iba1, and Cd68 as well as a new analysis of microglial marker expression in our transcriptomic dataset. Along with our in situ hybridization results, we find no changes in Rag2-/- microglia on the RNA or protein levels. Page 15, Fig S4.

Reviewer #1:

The authors set out to investigate whether in the healthy brain, lymphocytes play a role in the transcriptomic phenotype of glial cells. The study has solid foundation in the idea that adaptive immune changes correlate with changes to glial cells in various disease states. To answer the question the authors use lymphocyte (Rag2-/-) deficient mice in naïve states as well as control mice and conduct RNA sequencing on acutely purified microglia, astrocytes, OPCs, and oligodendrocytes. The sequencing reveals that once lymphocytes are eliminated there are no transcriptomic differences in glia except for modest differences seen in the oligodendrocytes, though gross morphology of WM remains the same. Although there is space and need for a publication of this kind, as it stands the publication is lacking clarity as to the overall impact of the findings and the authors must expand on the topic.

We thank the reviewer for acknowledging the need for a publication to address the impact of lymphocyte deficiency on glia. To clarify the impact of our findings, tandem with input from Reviewer 2, we took several steps to focus our narrative. First, we included a new finding: localization of the RNA-binding protein Quaking changes in oligodendrocytes, which provides a concrete launching point for further mechanistic study (Pg 13-14, Fig 2B). Second, we included new data that identifies sex-associated differences in gene expression in all four glial cell classes we examined in this study (Pg 15, S3 File). Third, we added context to our study by including a new paragraph in the introduction to demonstrate the widely studied roles of lymphocytes in neurological disease, which stands in contrast to the little that is known about their roles in homeostatic brain function (Pg 4). We also expand our description of lymphocytic impacts on behavior in the introduction to demonstrate the need to identify the molecular mechanisms underlying these changes (Pg 5). 

Synopsis: The authors set out to investigate whether in the healthy brain, lymphocytes play a role in the transcriptomic phenotype of glial cells. The study has solid foundation in the idea that adaptive immune changes correlate with changes to glial cells in various disease states. To answer the question the authors use lymphocyte (Rag2-/-) deficient mice in naïve states as well as control mice and conduct RNA sequencing on acutely purified microglia, astrocytes, OPCs, and oligodendrocytes. The sequencing reveals that once lymphocytes are eliminated there are no transcriptomic differences in glia except for modest differences seen in the oligodendrocytes, though gross morphology of WM remains the same. Although there is space and need for a publication of this kind, as it stands the publication is lacking clarity as to the overall impact of the findings and the authors must expand on the topic (see below).

• Given there were no differences or very slight differences found (in the case of the OLs) all analysis possibilities should be exhausted. 

We thank the reviewer for encouraging us to further expand upon our original data. We performed gene ontology term enrichment analysis, protein-protein interaction network analysis, and Gene Set Enrichment Analysis of our oligodendrocyte RNAseq data and did not identify enriched gene ontology terms. On the other hand, as mentioned above, we have now identified a protein-level difference in Rag2-/- oligodendrocytes that could have repercussions for RNA transport of genes that encode key myelin proteins (Pg 13-14, Fig 2B) [2, 3]. Per the reviewer’s next suggestion, we also leveraged our existing sequencing data to identify sex-associated gene expression in all four glial cell types (Pg 15, S3 File). 

• Authors provide numbers of mice in each group but not specific numbers of male and female mice used. Was the transcriptomic data striated by sex? Did this elucidate any differences? 

We thank the author for this actionable suggestion for generating new findings from our existing data. We now include differential gene expression analysis that identifies small to modest amounts of sex-associated gene expression in all four cell types (Page 15, S3 File). We found an unusually long list of sex-associated genes among oligodendrocytes, whose sexual dimorphism is not widely recognized at the transcriptomic level.

• Further, the immunopanning for oligodendroglia is conducted using two antibodies for OPCs and one for Mature OLs. Although O4 classically marks “OPCs”, it more specifically marks the stage of the lineage from committed OPCs to pre-myelinating OLs. Have the Authors tried separating the O4 and PDGFRα panned cells? Did this make a difference transcriptomically? 

We agree with the reviewer that there are important differences in selecting OPCs using PDGFRα vs. O4. We only collected OPC RNA from PDGFRα panned cells to avoid any contamination from immature oligodendrocytes. O4 plates were only used to further deplete OPCs and immature oligodendrocytes before collecting other cell types. We added language to the Methods and Results section to make our study design more explicit (Pg 6, 12).

• Figure 2A: Could the authors run a PCA on the data to get an overall view of gene expression changes between groups? 

We thank the reviewer for their suggestion to include more information for the reader. We have now included PCA plots for all four cell types in the supplemental materials (S5 Figure). However, we do not see obvious grouping by genotype when we plot the first two PCs of each cell type.

• Figure 2B: where were the images taken (more specifically)? Bregma? Were other myelin-rich regions checked? Can the authors include a quantification?

We thank the reviewer for suggesting a more thorough examination of MBP in the Rag2 model. We now include quantification of MBP immunostaining in three white matter rich regions (corpus callosum, anterior commissure, and striatum) (Pg 14, Fig 3A). In response to comments from Reviewer 2, we also performed further quantification of MBP and another myelin protein, PLP, using western blots (Pg 14, Fig 3B). Across all these analyses, we continue to find no differences between Rag2-/- and control mice. We clarified the location where the images were taken, including distance to Bregma (Line 224). 

• Lines 248-259: Microglial “maturation” occurs during developing stages when primitive macrophages progenitors migrate to the developing neural tube and become microglia. From that point on microglia clonally expand in response to insult or injury. The use of “maturation” when referring to microglia makes little sense, it must be clarified. 

We thank the reviewer for their suggestion. We agree that the vast majority of microglial maturation occurs in earlier embryonic stages though changes in microglial gene signatures can still be observed after birth. The language of “maturation” was borrowed from the dataset we used to compare MHCII-/- microglia[4]. To avoid confusion, we have eliminated all references to microglia “maturation”, and we simply refer to the “gene signature” of the MHCII-/- microglia.

• Figure 3A: if the authors had already shown no difference between RAG negative and positive mice in the RNA seq of microglia, there is little need for the RNA scope to be included as a main figure. 

We thank the reviewer for identifying an opportunity to focus our narrative by moving negative data into supplemental materials. We now include all comparisons of Rag2 microglia in a new supplemental figure, Figure S4. 

• Lines 254-259: Microglia are antigen-presenting cells that express MHCII. The comparison between their Rag2 negative and a lymphocyte-deficient MHCII KO model (Pasciuto) seems weird as one would affect microglia and the other wouldn’t. Please explain rationale.

We agree with the reviewer that the use of an MHCII KO could directly impact microglia. We edited the text to clarify the rationale of comparing our results with a previous report (Pg 17). Previously, Pasciuto et al. published a paper concluding that microglia require lymphocytes to complete fetal-to-adult transition (maturation) using MHCII KO mice[4]. Here, we wanted to assess whether our results are consistent with the previously published statement that microglia require lymphocytes to mature. We found that microglia development is normal in the absence of all lymphocytes. The differences between our findings and previous reports could be caused by a direct role of MHC II in microglia development or the differences in the complement of immune cells between Rag2 and MHCII models. We have amended the Discussion to include these important points (Pg 20).

• Lines 223-245: Were these upregulations confirmed via RNA scope or even immunofluorescence? Are these upregulations physiologically significant given gross myelin patterns appear normal? Please refer to comment on Figure 2B: can the authors provide further staining of myelinated regions w/ quantifications?

We thank the reviewer for raising the important point of validating our RNAseq findings in oligodendrocytes in order to determine the true biological relevance of these findings. After several attempts at qPCR validation, we were unable to get quantitative results (Cq values in the upper 30’s) due to low RNA concentrations in our immunopanning purified oligodendrocyte samples. Our RNAseq data was obtained using a kit specifically designed to amplify low concentrations of RNA (Nugen Ovation V2). Instead, we focused on pursuing protein-level differences in Rag2 mice, resulting in our finding that the RNA-binding protein Quaking is differently localized in Rag2-/- mice compared with controls. Quaking is known to bind MBP mRNA, which could have implications for myelination[2, 3]. As we described in the response to your previous point, we do not see evidence that the amount of myelin is changed in Rag2-/- mice. 

• Lines 272-282: In the GSEA the authors are again comparing a model which involves knocking out a microglial protein (MHCII) to their model which leaves microglia intact. Please explain rationale. 

We agree this is an important point to clarify. We have expanded our comparison of MHCII and Rag2 models in the Discussion, as we detail in our response to the reviewer’s previous comment above (Pg 20).

Reviewer #2: In this study, the authors sought to investigate whether lymphocytes are regulating the molecular phenotypes of glial cells (OPCs, oligodendrocytes, microglia, and astrocytes) in the healthy brain. They used a Rag2-/- mouse model (which results to the loss of all mature lymphocytes, including CD4+ T cells, CD8+ T cells, and B cells) to characterize the molecular phenotype of glial cells through RNA sequencing. The authors reported no changes in the molecular signatures of the majority of the glial cells, with the exception of oligodendrocytes. Considering the emerging interest on neuroimmune interactions (either innate or adaptive immune system) and the critical role they have upon the brain homeostasis and behavior, this study will help to further delve into potential connections between lymphocytes and glial cells in the CNS.

We thank the reviewer for acknowledging the impact of this study through its contribution to the rapidly growing interest in neuroimmune interactions. 

However, there are some major concerns as well as minor suggestions about the study:

1. The authors in the introduction section (and specifically lines 44-52) focus on studies related to meningeal lymphocyte roles in the brain. It is understandable that they should be distinguished from the peripherally recruited lymphocytes, however there is no other reference of the meningeal lymphocytes again in the manuscript and no distinction of them in the experiments they performed. Is there a specific reason the authors are underscoring this lymphocytic population? If not, then I would recommend using this space for the following suggestions (see end of comment 1).

We thank the reviewer for identifying a point that required further clarification in our text. We raise the notion of lymphocytes that specifically populate the meninges because the existence of these specialized cells in proximity to the brain suggests that they may have significant interactions that impact brain function. This directly speaks to the importance of this study, namely to demonstrate the impact (or lack thereof) that peripheral lymphocytes exert on brain cells. We also include citations that describe a molecular mechanism whereby meningeal lymphocytes regulate behavior via neuronal signaling; our manuscript seeks to identify whether lymphocytes can exert influence on other brain cells, i.e. glia. We added text to the Introduction to reflect this reasoning (Pg 3). If the reviewer feels that introducing meningeal lymphocyte distract from the main message of the manuscript, we can remove these sentences.

Moreover, in the next paragraph (lines 56-72), the authors provide a general background about functions of glial cells in health and disease. I agree that it is important to introduce the glial cells, since it is an important aspect of this study, however It should not be narrated as a glial section from a review paper. My recommendation would be to focus on the neuroimmune interactions and the effects they have on glial cells and brain homeostasis, and importantly provide more information about the roles of lymphocytes in CNS health and disease-since this is the central scope of the paper (should be significantly expanded more than just the 81-83 lines).

We thank the reviewer for identifying another opportunity to sharpen the narrative of our manuscript. In the Introduction, we have reduced the general background on glia by eliminating background that specifically covers the functions of astrocytes and OPCs, as they are not the focus of this study. In addition, we include an entirely new paragraph that describes the involvement of lymphocytes in neurological disease, in order to demonstrate the extent to which these cells participate in brain function under pathological conditions (Pg 4). This stands in contrast to the paucity of understanding of lymphocytic roles during homeostasis, particularly regarding glia. 

 Along the same lines the authors should provide more information about the Rag2-/- animal model in the introductory section, and specifically discuss previous findings regarding the cellular, molecular and behavioral mechanisms shown in the literature (i.e. refs 61-62). The reader should be able to understand the current gap in knowledge and how this paper seeks to contribute on that.

We significantly expanded the description of these previous studies in the Introduction to more concretely illustrate the regulation of lymphocytes on animal behavior (Pg 5).

2. The authors mention in lines 92-94: «Overall, we find little evidence that lymphocytes influence CNS function by majorly contributing to the cellular states of glial cell types in the healthy brain». I find this inaccurate for 3 reasons: i) this study focuses only in tissue isolated from cerebral cortex (excluding the remaining brain areas), 

ii) the RNA sequencing analysis portrays molecular signatures and not cellular states of glial cells, and iii) the molecular changes found in the oligodendrocyte populations should have been further characterized to be able to conclude on that (as later will be discussed).

We thank the reviewer for the insightful point and modified the sentence to be more accurate. Now it reads “Overall, we find little evidence that lymphocytes influence CNS function by majorly altering the transcriptome profiles of microglia, astrocytes, and OPCs in the healthy cortex” (Lines 115-117). As we will address in response to later points, we now include a protein-level difference that further validates the impact of lymphocytes on oligodendrocyte populations. 

3. In the materials & methods section (line 116), the anti-CD45 also targets a small population of resident macrophages (~1-3%). This should be depicted in the manuscript.

We thank the reviewer for this important clarification. We have added this caveat to our Methods (Lines 145-146).

4. With regard to Figure 1 data the authors:

- Have not included information regarding either in figure legend or in the methods what z-score depicts and based on which control group the heatmap scale was made (increase or reduction of TPM compared to what control).

Z-scores were defined as (expression in the sample - the average expression across all samples)/standard deviation. This language has been added to the figure legend (Line 267).

- I would recommend including more marker genes (at least 4 more classic markers) for each glial cluster.

We have added 4 markers (7-8 markers total) for each cell type to the heatmap to better demonstrate the purity of our samples (Fig 1C). This did not change our conclusion that we obtained highly pure samples, with the exception of oligodendrocyte contamination in the astrocyte samples. 

- I would recommend the authors to provide a dot-plot analysis displaying the average expression levels (Change to avg. exp. scale), as well as the percentage of cells within each cell cluster expressing each marker gene (% Expression), split in groups of Rag2-/- vs Rag2+/+ mice. This information is needed to grasp the molecular signature of each group as well as whether the % of cells expressing the markers changes. 

This study utilizes bulk RNA sequencing, and we therefore do not have expression data from individual cells, which means we are unable to calculate the percent of cells expressing a given gene. What we plot in Figure 1 is the maximum level of granularity, as each sample only has one data point per gene.

5. As in Figure 1, in Figure 2 more information about the volcano-plots should be included in the figure legend and the material & methods section.

We have expanded the text describing volcano-plots used in Figure 2 (Line 296). 

6. The authors demonstrate an important finding depicting that lymphocyte deficiency dramatically affects the oligodendrocyte transcriptome. However, instead of focusing on this novel finding and try to strengthen their hypothesis with supplementary experiments, they only use one and a half panel [2A (Oligodendrocytes) and 2B] and they conclude in lines 244-245 that: «Despite the change in gene expression, gross patterns of myelin appear unchanged in Rag2-/- mice based on immunofluorescence of myelin basic protein (Fig 2B)».

Therefore, there are some major points that the authors should improve for this section:

i) Perform RNA expression analysis (i.e. RT-qPCR) to validate some things that showed up in their RNA sequencing analysis.

We agree that further validation of RNAseq data is ideal. Unfortunately, our immunopanning purified oligodendrocyte RNA samples had concentrations too low to obtain reliable qPCR data when we attempted to validate several differentially expressed genes. Our RNAseq pipeline had been optimized for very low input through the use of a kit specifically designed for that purpose. In light of this, we refocused our energies on protein-level differences in Rag2-/- oligodendrocytes, and we found these cells alter the localization of the RNA-binding protein, Quaking (Pg 13-14, Fig 2B). Quaking protein isoform 7 is detected in the soma and processes of control oligodendrocytes but only in the soma of Rag2-/- oligodendrocytes. Quaking is known to bind the key myelin transcript MBP. This observation is the first report of a lymphocyte-dependent subcellular localization change of an oligodendrocyte protein, to the best of our knowledge.

ii) Then the next step would be to check protein expression (i.e. immunoblot analysis of some classic myelin or generally mature oligodendrocyte markers).

We thank the reviewer for this actionable suggestion. We performed western blots to measure the levels of two major myelin proteins, MBP and PLP (Fig 3B). As with our immunostaining, we did not find evidence that the protein level of these myelin markers differs in Rag2-/- mice. 

iii) The authors provide 2 representative immunofluorescent staining for MBP, which brings up several issues:

- There are no graphs or statistics to support their claims

- These images actually depict a decrease in Rag2-/- mice

- But even if the representative images are wrongly selected, there is no information which area of the cortex is this. The white matter can be dramatically different depending on the area and the bregma coordinates these sections are from.

- The reader should be able to appreciate a larger cortical area (use lower magnification image and include insets with higher magnification)

- I would strongly recommend quantification, if there has not been performed already, which should be normalized per area (or use integrated analysis).

We now include quantification of MBP immunostaining and new representative images that demonstrate a more comprehensive view of the MBP signal (Fig 3A). We quantified MBP signal from three myelin-rich regions: corpus callosum, anterior commissure, and striatum. Consistent with our original observation and new western blot data, we do not detect any differences between Rag2-/- and control mice. Information on bregma is also included in the Methods (Line 224). 

7. As mentioned in comment 6, the authors decided to not follow the innovative results they had from the RNA sequencing (Figs 1 and 2), but instead they used the whole Figure 3 to compare their findings on microglia with the previous publication of Pasciuto et al., Cell (2020). In my opinion the purpose and the flow of experiments in a study should not be determined by another study but on the hypothesis the authors have. On many occasions (lines 254-259) in the results section the authors compared their results with the previous study (it is more usual to do so in the discussion section), feeling as the sole purpose of this study was to prove the Pasciuto publication wrong. I would have been a lot more supportive on the narrative that the authors decided to take in Figure 3, if there was conclusive data of no effects of Lymphocytes upon the Oligodendrocytes (and as a consequence the authors sought investigate the microglia in more depth). However, the authors performed only a superficial characterization of Oligodendrocytes, and therefore decided to neglect the novel findings of their RNAseq study, making hard to follow the exact hypothesis of the study.

We thank the reviewer for identifying opportunities to focus the narrative of our study. As mentioned above, we expanded our analyses of oligodendrocyte proteins and identified an interesting subcellular localization difference in Quaking in Rag2-/- vs. control mice (Pg 13-14, Fig 2B) and no change in MBP or PLP proteins (Fig 3B). We edited the text to reemphasize our oligodendrocyte results and clarified the rationale for including the comparison with the Pasciuto publication (Pg 17). We have deemphasized our microglial results by moving nearly all negative data describing microglia in our study into a new supplemental figure (Figure S4), and we have expanded the Discussion to elaborate on a unifying hypothesis consistent with the results of both studies and points to future directions of exploration (Pg 20).

8. Regarding Figure 3, as previously mentioned in comment 6, the authors should also include either in the figure legend or the materials section, the following information:

- How the quantification of RNAscope analysis was performed.

- Which area of the cortex was analyzed

- It should also be depicted on the y axis of the graphs that the mean fluorescent intensity was normalized to the area.

We have added text to the Methods to better describe our RNAscope experiments, including specifying the regions examined (both upper and lower cortex in the motor and somatosensory regions, Lines 199-201). For clarification, we did not explicitly normalize the mean fluorescence intensity to the area, that is included by definition in “mean fluorescence”, as “mean” signifies that individual fluorescence values were averaged across all pixels, which are themselves a measure of area. We did separately quantify signal area, but that is an entirely different measurement; our fluorescence measure assessed the signal across the whole image, and our area measure assessed what proportion of the image showed fluorescence that passed a given threshold. 

9. The authors in the first sentence of the discussion (lines 313-314): In this study, we find that microglia in adult Rag2-/- mice under homeostatic conditions are indistinguishable from microglia in immunocompetent mice. However, apart from the RNA sequencing data and the RNA expression of Tmem119 and C1qa, there is no other support for the “indistinguishable” phenotype the authors claim. I would recommend characterization of supplementary microglial markers not only on a RNA level (some classic markers for RT-qPCR: P2RY12, PTPRC, CX3CR1, CTSS, LPAR6, CD68, ARHGAP24, ITGAM, AIF1), but most importantly for protein expression experiments. In my point of view, what the authors depict on this study is that there is only an indication of no substantial molecular signature differences in microglia during lymphocyte deficiency, which remains to be further examined with the aforementioned experiments (especially when comparing these results with the Pasciuto et al. publication, which has dedicated a large palette of experimental approaches to conclude to their findings).

We performed a new analysis and a new experiment to more thoroughly examine microglia in Rag2-/- and control mice. First (and in reference to the reviewer’s next comment), we returned to our RNAseq data to extract the expression of a large panel of microglial genes and created a general score of microglial gene expression, which showed no differences between groups (Fig S4C). Second, we performed immunostaining of three highly expressed microglial proteins, Iba1, P2ry12, and Cd68. Once again, we saw no differences in the Rag2-/- mice compared with controls (Fig S4D). Together, we now show microglia do not change at the RNA level (in bulk sequencing or in situ) or the protein level, using a variety of microglial markers. We also edited the text to more accurately describe our findings. It now reads “we find that microglia in adult Rag2-/- mice under homeostatic conditions are indistinguishable from microglia in immunocompetent mice in their transcriptome profiles” (Line 457).

10. Furthermore, in Fig.3a-b based on the RNAscope in situ hybridization in Rag2-/- and Rag2+/+ brains, the authors conclude that the maturation of microglia is unaffected by the lymphocyte deficiency. However, again this cannot be concluded just by fluorescent quantification of just three RNA expression markers, and no protein analysis. I would at least request the authors to utilize their RNAseq expression dataset to complement these findings with a comparative transcriptional analysis using a wide range of microglial maturation genes from Rag2-/- vs the Rag2+/+ mice, in order to get a general score.

We thank the reviewer for this suggestion to generate further insight from the data we have already collected. We created a panel of 12 genes highly expressed in microglia, then we converted the gene expression (TPM values) to z-scores, based on the mean expression across all samples (Fig S4C). Importantly, to avoid bias, we did not iteratively test various panels of genes; we compiled a single panel of genes that were highly expressed in microglia using an independent, previously published RNAseq of glia. We then created a single microglia expression score by taking the average z-score for each sample. In supplemental figure S4C, we show a heatmap depicting all genes for each sample. Reassuringly, we observe similar z-scores within each individual sample. When we plot our combined expression scores, we do not find differences between genotypes.

11. Of importance, a clarification is required for the Fig.3c. The authors took all the differentially expressed genes in microglia from the MHCII-knockout mice from the Pasciuto study and performed gene set enrichment analysis (GSEA) using their RNA-seq data. Based on that the authors found that the up- and down-regulated genes identified in this study did not show global enrichment in this dataset. The question is, was the whole cluster of microglia used for this GSEA analysis or just the microglial subcluster 3 that is mentioned in the Pasciuto study?

We reference the Pasciuto analysis that compared all MHCII-/- microglia to all control microglia. We have added text to specify which comparison we are using in our analysis (Lines 180-181).

12. In the discussion the authors suggest possible explanations of the different results shown on this study compared to the Pasciuto et al., publication. To this end, it would be important to further describe the differences between the Rag2 KO mice and the MHCII KO used on the other study. This way the reader would be able to appreciate the findings of this study and the deviations between the studies.

We thank the reviewer for encouraging us to provide a more thorough explanation of this key point in order to fully demonstrate the importance of our study. We expanded this section of our Discussion describing these two models, and we go into greater detail on the conclusions we have drawn based on their distinct outcomes (Pg 20).

13. The authors should include a statistics section in the methods. Also, summary statistics, the data points behind means, medians and variance measures should be available.

We thank the reviewer for ensuring greater transparency in our dataset. We now include the group means and standard deviation for each experiment along with the exact p-value, either in the Results or the associated figure legend. We also add a statistics section to the Methods (Pg 11).

 

References

1. Bin JM, Harris SN, Kennedy TE. The oligodendrocyte-specific antibody 'CC1' binds Quaking 7. J Neurochem. 2016;139(2):181-6. Epub 2016/07/28. doi: 10.1111/jnc.13745. PubMed PMID: 27454326.

2. Larocque D, Pilotte J, Chen T, Cloutier F, Massie B, Pedraza L, et al. Nuclear retention of MBP mRNAs in the quaking viable mice. Neuron. 2002;36(5):815-29. Epub 2002/12/07. doi: 10.1016/s0896-6273(02)01055-3. PubMed PMID: 12467586.

3. Li Z, Zhang Y, Li D, Feng Y. Destabilization and mislocalization of myelin basic protein mRNAs in quaking dysmyelination lacking the QKI RNA-binding proteins. J Neurosci. 2000;20(13):4944-53. Epub 2000/06/24. PubMed PMID: 10864952; PubMed Central PMCID: PMCPMC6772302.

4. Pasciuto E, Burton OT, Roca CP, Lagou V, Rajan WD, Theys T, et al. Microglia Require CD4 T Cells to Complete the Fetal-to-Adult Transition. Cell. 2020;182(3):625-40 e24. Epub 2020/07/24. doi: 10.1016/j.cell.2020.06.026. PubMed PMID: 32702313.

---

## [Decision Letter · Decision Letter 1]

14 Dec 2022

Lymphocyte deficiency alters the transcriptomes of oligodendrocytes, but not astrocytes or microglia

PONE-D-22-23915R1

Dear Dr. Zhang,

We’re pleased to inform you that your manuscript has been judged scientifically suitable for publication and will be formally accepted for publication once it meets all outstanding technical requirements.

Kind regards,

Stella E. Tsirka

Academic Editor

PLOS ONE

Additional Editor Comments (optional):

Reviewers' comments:

Reviewer's Responses to Questions

**Comments to the Author**

1. If the authors have adequately addressed your comments raised in a previous round of review and you feel that this manuscript is now acceptable for publication, you may indicate that here to bypass the “Comments to the Author” section, enter your conflict of interest statement in the “Confidential to Editor” section, and submit your "Accept" recommendation.

Reviewer #1: All comments have been addressed

2. Is the manuscript technically sound, and do the data support the conclusions?

Reviewer #1: Yes

3. Has the statistical analysis been performed appropriately and rigorously? 

Reviewer #1: Yes

4. Have the authors made all data underlying the findings in their manuscript fully available?

Reviewer #1: Yes

5. Is the manuscript presented in an intelligible fashion and written in standard English?

Reviewer #1: Yes

6. Review Comments to the Author

Reviewer #1: The authors adequately responded to all questions provided. The paper was scientifically sound to begin with, but required some clarifications in wording and a few extra experiments.

7. PLOS authors have the option to publish the peer review history of their article (what does this mean?). If published, this will include your full peer review and any attached files.

Reviewer #1: No

---

## [Editor Report · Acceptance letter]

15 Feb 2023

PONE-D-22-23915R1 

Lymphocyte deficiency alters the transcriptomes of oligodendrocytes, but not astrocytes or microglia 

Dear Dr. Zhang:

I'm pleased to inform you that your manuscript has been deemed suitable for publication in PLOS ONE. Congratulations! Your manuscript is now with our production department. 

Kind regards, 

on behalf of

Dr. Stella E. Tsirka 

Academic Editor

PLOS ONE